# Skin Barrier Function and Microtopography in Patients with Atopic Dermatitis

**DOI:** 10.3390/jcm13195861

**Published:** 2024-10-01

**Authors:** Carlota Pretel-Lara, Raquel Sanabria-de la Torre, Salvador Arias-Santiago, Trinidad Montero-Vilchez

**Affiliations:** 1Dermatology Department, School of Medicine, University of Granada, 18016 Granada, Spain or carlotapretel@correo.ugr.es (C.P.-L.); or tmonterov@ugr.es (T.M.-V.); 2Instituto de Investigación Biosanitaria ibs GRANADA, 18012 Granada, Spain; raquelsanabriadlt@gmail.com; 3Dermatology Department, Virgen de las Nieves University Hospital, 18014 Granada, Spain; 4Department of Biochemistry, Molecular Biology III and Immunology, University of Granada, 18071 Granada, Spain

**Keywords:** atopic dermatitis, eczema, homeostasis, skin barrier, visioscan

## Abstract

**Background**: Atopic dermatitis (AD) is a chronic inflammatory skin disease whose incidence is increasing. Skin barrier dysfunction plays an important role in this disease. It has been observed that AD patients have higher transepidermal water loss (TEWL) and lower stratum corneum hydration (SCH); however, there is little information about skin microtopography in this pathology. The objective of this study is to evaluate skin barrier dysfunction and structural changes in patients with AD. **Methods**: A cross-sectional study was conducted including patients with AD. Parameters of skin barrier function were measured (TEWL, temperature, erythema, pH, skin hydration, elasticity) and also other topographical parameters (scaliness, wrinkles, smoothness, surface, contrast, variance) in both healthy skin and flexural eczematous lesions. **Results**: A total of 32 patients with AD were included in the study. Flexural eczematous lesions had higher erythema (369.12 arbitrary unit (AU) vs. 223.89 AU, *p* < 0.001), higher TEWL (27.24 g/h/m^2^ vs. 13.51 g/h/m^2^, *p* < 0.001), lower SCH (20.3 AU vs. 31.88 AU, *p* < 0.001) and lower elasticity (0.56% vs. 0.65%, *p* = 0.05). Regarding topographic parameters, flexural eczematous lesions presented greater scaliness (5.57 SEsc vs. 0.29 SEsc, *p* = 0.02), greater smoothness (316.98 SEsm vs. 220.95 SEsm *p* < 0.001), more wrinkles (73.33 SEw vs. 62.15 SEw *p* = 0.03), greater surface area (836.14% vs. 696.31%. *p* < 0.001), greater contrast (2.02 AU vs. 1.31 AU *p* = 0.01), greater variance (6.22 AU vs. 4.96 AU *p* < 0.001) and a lower number of cells (105.5 vs. 132.5 *p* < 0.001) compared to unaffected healthy skin, reflecting a decrease in skin quality in AD patients. **Conclusions**: Both skin barrier function and skin topography are damaged in patients with AD, with differences between healthy skin and flexural eczema.

## 1. Introduction

Atopic dermatitis (AD) is one of the most common chronic inflammatory skin diseases. Recent data suggest that the prevalence among children is approximately 20% and among adults ranges from 7% to 14%, with variation between countries and higher prevalence in industrialised countries [1]. It is a chronic disease that flares with inflamed, red, itchy plaques [2]. Other signs include oedema, xerosis, erythema, oozing, erosion, crusting and lichenification [3]. A relevant clinical feature is the presence of very dry skin [4].

The skin is the largest organ of the human body [5]. Its main functions are to regulate temperature, both by isolation and sweating; to participate in nervous system functioning and water content regulation; and to protect the body from mechanical injury, micro-organisms, substances and radiation present in the environment [6]. It also acts as a physical, chemical and immunological barrier between the body and the environment [7]. This function resides mainly in the stratum corneum of the epidermis. It is important to highlight the commensal microbiome in this barrier, which due to aggression from various internal and external factors can lead to inflammation, infection or allergic or autoimmune skin diseases. Genetic predisposition, immune system dysregulation and epidermal barrier dysfunction are some of the crucial components of AD [8,9].

Patients with AD have a reduction in total ceramide levels and an alteration in their profile, hence the importance of applying topical ceramides in this condition [4]. Filaggrin, a binding protein that holds corneocytes together, becomes relevant as it helps to maintain transepidermal water loss (TEWL) within normal values [10]. Thus, factors contributing to impair the skin include decreased or mutated filaggrins [11], lipids and antimicrobial peptides.

A number of parameters such as TEWL, stratum corneum hydration (SCH), temperature, skin surface pH, erythema index, elasticity and antioxidant capacity are usually involved in determining and measuring barrier function [8,12]. Previous research on AD has shown that patients with AD have increased TEWL, erythema and temperature, as well as reduced SCH and elasticity on flexural eczematous lesions [4,12]. In addition, more severe patients had greater dysfunction of these parameters. Moreover, early changes in theses parameters could predict treatment response. Nevertheless, there is little information about skin microtopography in patients with AD.

Therefore, the aim of this study was to evaluate epidermal barrier dysfunction and skin microtopography in patients with AD.

## 2. Materials and Methods

Study design

A cross-sectional study was designed with patients diagnosed with AD attending the Atopic Dermatitis Clinic of the Hospital Universitario Virgen de las Nieves, Granada, Spain, who voluntarily wished to participate and who met the inclusion criteria and none of the exclusion criteria.

Inclusion and exclusion criteria

The inclusion criteria were as follows:–Patients with AD diagnosed by a dermatologist following the Hanifin and Rajka criteria of at least 6 months of evolution and who have attended the Atopic Dermatitis consultation at the Dermatology Department of the Hospital Universitario Virgen de las Nieves–Aged between 12 and 65 years of age–Have signed the informed consent required to participate in the study

Exclusion criteria include the following:–Subjects who do not wish to participate in the study and who do not sign the informed consent form–Patients with clinical infection in the area to be measured–History of cancer or immunosuppression–Any concomitant inflammatory skin pathology (psoriasis, hidradenitis suppurativa)

Variables of interest

Skin barrier function

The biophysical and objective parameters concerning the assessment of the epidermal barrier function were measured using probes with a multiprobe adapter as a non-invasive test. These measurements included temperature (°C) with the Skin-Thermometer ST 500, erythema (UA) with the Mexameter, TEWL (g/h/m^2^) with the Tewameter, stratum corneum hydration (SCH) (UA) with the Corneometer, pH of the skin surface with the Skin pH-Meter, skin elasticity (%) with the Cutometer, skin hardness (UA) with the Durometer, skin friction (UA) with the Frictiometer, skin gloss (UA) with the Skin-Glossymeter, skin deformability or stiffness (mm) with the Indentometer and skin antioxidant capacity (UA).

Skin microtopography

The surface evaluation of living skin (SELS) parameters smoothness (SEsm), roughness (SEr), scaliness (SEsc) and wrinkles (SEw) were assessed using the Visioscan VC 20plus (Courage + Khazaka electronic GmbH, Bilbao, Spain), high-resolution UVA light video camera. Lower SEsm means higher smoothness, lower SEr is related to higher roughness, lower SEsc means less desquamation and higher SEw is related to more wrinkles. The surface area (%), volume (mm^2^), contrast (UA), entropy (UA), variance (UA), homogeneity (AU) and anisotropy (AU) were also measured using the Visioscan. Entropy indicates ‘image disorganisation/disorder’ concluding that smooth and regular skin should show a lower entropy than rough skin. Variance expresses roughness, so high roughness will lead to increased variance values. High homogeneity is related to moisturised skin and high anisotropy means more aged skin. The total number of cells, the rate of desquamation and the number of corneocytes were measured using the Corneofix.

Skin barrier function and skin microtopography measurements were carried out in a room at a temperature of 23 +/−1 degree and an ambient humidity in the range of 40–50%. They were measured on a flexural eczematous lesion on the volar forearm and an uninvolved area 3 cm from the lesion.

Severity and quality of life

To evaluate the severity of the disease, we used the following scales:°Scoring Atopic Dermatitis (SCORAD), an index that evaluates the extent with a percentage of the affected surface area, the intensity of AD through six items (erythema, oedema/papules, effect of scratching, oozing/crusting, lichenification and dryness) scored from 0 (absence) to 3 (intense) and two subjective symptoms, itching and insomnia, scored from 0 to 10. The total is calculated = extent/5 + 7 × intensity/2 + symptoms of itching and insomnia. The disease is considered mild if it has a score between 0 and 25; moderate between 25 and 50; and severe >50.°The Redness, Oedema, Scratches (ROS) scale which assesses erythema, oedema and excoriations. Each item is scored from 0 to 3 and the total score, adding up each item, will give a number between 0 and 9.°The Eczema Area and Severity Index (EASI) scale, that includes the extent of disease and the percentage of body surface area in four body regions (head and neck, trunk, upper limbs and lower limbs). The disease is mild if EASI ≤ 7, moderate 7.1–21, severe >21–50 and very severe >50.°The Investigator’s Global Assessment (IGA) which uses the clinical features of erythema, infiltration, papules, oozing and crusting to classify it from 0 (clear) to 5 (very severe disease).

Patients were also provided with a series of questionnaires related to their experience with this condition. The first was the Dermatology Life Quality Index (DLQI) as an AD monitoring tool scored between 0 and 24, with a number over 7 indicating poor control. The second was the Patient-Oriented Eczema Measure (POEM) to collect patient-reported symptoms, consisting of seven questions, each weighted from 0 to 4; the score is summed and expressed as a maximum of 28. Two others were the Numerical Rating Scale (NRS) for pruritus rated from 0 (no pruritus) to 10 (most intense pruritus the patient can imagine) and the NRS for sleep, which assesses how the pathology affects sleep, also rated on a scale of 0 to 10.

Other variables

Sociodemographic variables (sex, age, occupation, level of education, residence), toxic habits, phototype, skin care in terms of moisturiser use, sun exposure and photoprotection were included. We also collected information on whether there was a family history of AD. With regard to the pathology in question, we recorded the age of onset of AD, comorbidities (triad of asthma, rhinoconjunctivitis and AD), location of the lesions, current treatment for AD and previous treatments.

Statistics

In a descriptive analysis, continuous variables were expressed as means ± standard deviations (SD) and qualitative variables as absolute and relative frequency distributions. For comparisons of continuous variables, Student’s t-test for independent samples or Student’s t-test for paired samples, as appropriate, was used. For comparisons between categorical variables, the chi-square or Fisher’s test was used as appropriate. Pearson’s correlation coefficient was calculated to test for possible correlations between continuous variables. Statistical significance was defined by a two-tailed *p* < 0.05. SPSS version 24.0 (SPSS Inc., Chicago, IL, USA) was used for statistical analyses.

Ethics

The study was conducted in accordance with the guidelines of the Declaration of Helsinki, and was approved by the Ethics Committee of the Hospital Universitario Virgen de las Nieves, Granada, Spain (code HC01/0442-N-20, impact of topical, systemic or physical treatment on skin homeostasis in patients with skin diseases (changes in skin homeostasis in dermatological patients)). Involvement in the study was explained to the participants, who decided to sign the informed consent and participate. The measurements performed on each patient were non-invasive and the confidentiality of personal data was preserved.

## 3. Results

### 3.1. Descriptive Analysis and Sociodemographic Characteristics of the Study Population

The study included thirty-two participants, of whom 37.5% (twelve) were male and 62.5% (twenty) female, with a mean age of 30.91 (14.29), Table 1. A total of 18.8% (six) smoked and 40.6% (thirteen) drank an average of 1.38 (2.08) Standard Beverage Units (SBU) per week. Regarding the use of moisturisers, 90.6% (twenty-nine) reported using cream. The mean age at onset of AD was 11.63 (19.56) years. At the time of inclusion in the study, 37.5% (twelve) of the patients were being treated with topical corticosteroids, 21.9% (seven) with a systemic drug, 18.8% (six) with a biologic and 21.9% (seven) with a JAK inhibitor (iJAK). In relation to disease severity, patients had moderate AD, reflected in a mean SCORAD of 37.37 (27.44) and an EASI of 13.54 (14.46).

### 3.2. Differences in Skin Barrier Function and Skin Microtopography between Healthy Skin and Flexural Eczematous Lesions

Skin barrier function and skin microtopography were different between healthy skin and flexural eczematous lesions (Table 2). Flexural eczematous lesions presented higher temperature (32.38 °C vs. 31.81 °C, *p* = 0.02), greater erythema (369.12 AU vs. 223.89 AU, *p* < 0.001), greater TEWL (27.24 g/h/m^2^ vs. 13.51, g/h/m^2^ *p* < 0.001), lower SCH (20.3 AU vs. 31.88 AU, *p* < 0.001), lower elasticity (0.56% vs. 0.65%, *p* = 0.05), higher hardness (15.75 AU vs. 9.63 AU, *p* = 0.02), lower friction (102.99 AU vs. 177.84 AU, *p* = 0.02), higher smoothness (316.98 SEsm vs. 220.95 SEsm, *p* < 0, 001), higher desquamation (5.57 SEsc vs. 0.29 SEsc *p* = 0.02), more wrinkles (73.33 SEw vs. 62.15 SEw *p* = 0.03), higher surface area (836.14% vs. 696.31% *p* < 0.001), higher contrast (2.02 AU vs. 1.31 AU, *p* = 0.01), lower entropy (1, 43 AU vs. 1.48 AU, *p* < 0.001), higher variance (6.22 AU vs. 4.96 AU *p* < 0.001), a lower total cell number (105.5 vs. 132.5, *p* < 0.001), a lower percentage of corneocytes (36.19% vs. 50.35%, *p* = 0.02) and a lower desquamation index (19.36 vs. 26.1, *p* = 0.04).

### 3.3. Differences between Men and Women

In an analysis grouped by sex (12 men and 20 women) (Appendix A), we found a higher mean age in men without this being significant (36.83 years vs. 27.35 years, *p* = 0.07). In addition, women used photoprotectors (65% vs. 25%, *p* = 0.03) and moisturisers more frequently (100% vs. 75%, *p* = 0.02). In terms of severity, men had a higher severity reflected in higher values for EASI (20.08 vs. 9.62, *p* = 0.05), IGA (2.67 vs. 1.65, *p* = 0.05) and NRS sleep (5.27 vs. 2.5, *p* = 0.05).

In barrier function parameters, men had higher erythema in both healthy skin (260.41 AU vs. 201.97 AU, *p* = 0.03) and flexural eczema (415.98 AU vs. 335.65 AU, *p* = 0.01). Friction in eczema was lower in males (80.04 AU vs. 119.39 AU, *p* = 0.04), deformability/stiffness in healthy skin was lower in males (1.83 mm vs. 2.11 mm *p* = 0.05) and anisotropy in flexural eczema was also lower in males (25.43 AU vs. 40.69 AU, *p* = 0.02).

### 3.4. The Impact of Age in Skin Barrier Function and Skin Microtopography

Patients were divided regarding the mean age (30.91) of the population (patients ≥31 and <31) (Appendix A). Patients aged ≥31 used moisturisers less frequently than those aged <31 (75% vs. 100% *p* = 0.02). Age at onset of AD was significantly higher in patients aged ≥31 (25.75 vs. 3.15, *p* = 0.01). Erythema in flexural eczema was higher in the population ≥31 years (422.54 AU vs. 347.12 AU, *p* = 0.03) and TEWL in flexural eczema was significantly higher (30.36 g/h/m^2^ vs. 19.66 g/h/m^2^, *p* = 0.04) in patients <31 years. SCH in healthy skin was higher in patients aged ≥31 (39.59 AU vs. 27.26 AU *p* < 0.001). Friction in healthy skin (227.9 AU vs. 147.81 AU, *p* = 0.05) and the healthy skin desquamation index (30.26 vs. 23.6, *p* = 0.05) were significantly higher in patients aged ≥31.

### 3.5. Differences in Skin Barrier Function and Skin Microtopography Depending on Disease Severity

Regarding disease severity, patients were divided into those having a severe AD (EASI ≥ 21) and those having a mild–moderate disease (EASI < 21) (Appendix A). Those with an EASI score ≥21 had significantly less sun exposure (20% vs. 63.6%, *p* = 0.02) and less sun exposure time (24 min vs. 77.73 min, *p* = 0.05). Patients with EASI ≥ 21 had more comorbidities than patients with EASI < 21 (80% vs. 31.8%, *p* = 0.01). In addition, those with EASI ≥ 21 had more severe and worse controlled disease reflected in higher scores on all AD rating scales (SCORAD, ROS, IGA, POEM, DLQI, NRS pruritus, NRS sleep). Erythema on healthy skin was also higher in patients with EASI ≥ 21 (288.31 AU vs. 194.6 AU, *p* = 0.01). The pH in flexural eczema was higher in the group with EASI ≥ 21 (5.81 vs. 5.02, *p* < 0.001). The R7 elasticity in flexural eczema was lower in subjects with EASI ≥ 21 (0.48% vs. 0.61%, *p* = 0.01) and so was the hardness in healthy skin (7.51 AU vs. 10.59 AU, *p* = 0.04). The brightness in flexural eczema was lower in those with EASI ≥ 21 (5.1 AU vs. 6.44 AU, *p* = 0.01). Finally, the total number of cells in flexural eczema was lower in patients with EASI ≥ 21 (84 vs. 118.4, *p* = 0.02).

### 3.6. Correlations between Severity Scales and Barrier Function and Topographical Parameters

Disease severity was related to changes in skin barrier function and skin microtopography (Table 3). A positive correlation was observed between EASI and erythema on healthy skin (r = 0.56, *p* < 0.001), pH on flexural eczematous lesions (r = 0.66, *p* < 0.001), smoothness on healthy skin (r = 0.4, *p* = 0.02), desquamation on healthy skin (r = 0.49, *p* = 0.04), contrast on healthy skin (r = 0.46, *p* = 0.01) and homogeneity on healthy skin (r = 0.46, *p* = 0.01), while a negative correlation was found between EASI and brightness on flexural eczematous lesion (r = −0.48, *p* = 0.02) and entropy on healthy skin (r = 0.42, *p* = 0.02) and on the total number of cells (r = −0.42, *p* = 0.04), Figure 1. Other correlations were also found between SCORAD, IGA, NRS for pruritus and sleep and skin barrier function and skin microtopography.

## 4. Discussion

This study shows that there are differences in barrier function and skin microtopography between healthy skin and flexural eczema in patients with AD and that sex, age and disease severity could impact on these parameters.

Concerning barrier function parameters, flexural eczema had higher temperature, higher erythema, higher TEWL, lower hydration and lower elasticity. These results are consistent with findings already described in previous studies [8,12,13,14,15]. The alteration of these parameters is evidence of an alteration in the barrier function of patients with AD; the increase in temperature demonstrates the inflammatory changes that occur in this pathology, as does the erythema. The increase in TEWL and the decrease in SCH reflects the dysfunction of the skin barrier due to the possible presence of mutations or alterations in the number of filaggrins, which are the main constituent proteins of the stratum corneum, which is mainly responsible for this barrier. Some studies have demonstrated the relationship between higher TEWL, greater erythema and the presence of mutations in these proteins [11]. Regarding elasticity, in the flexural eczematous lesions there was a decrease in elasticity, specifically in R7 (comparative elasticity or ratio between elastic recovery and total deformation of the skin), which may be due to an alteration in collagen and elastin, the proteins responsible for elasticity [8].

On the other hand, in our study, differences were observed between healthy skin and flexural eczematous lesions with respect to skin microtopography parameters. Flexural eczema showed an increase in the smoothness parameter (SEsm). Higher SEsm means a less smooth skin. More wrinkles (SEw) were also observed in flexural eczematous lesions, which means the flexural eczema has more visible wrinkles (wide and deep) and more desquamation (SEsc) in the stratum corneum. All these findings imply a poorer skin quality in patients with AD. These parameters have been little studied in previous studies in patients with AD. In healthy individuals, it was observed that skin barrier and microtopography differed between anatomical regions [16]. In patients with AD, the daily topical application of a nanoemulsion containing cholesterol derivatives of ceramide-like lipoamino acids during 6 weeks decreased scaliness and smoothness. In another study in children with AD, the population was divided into three groups; two received topical care with a plant-based emollient and the control group received petroleum-jelly-based emollient. After the follow-up, scaliness decreased in all the groups while smoothness only increased in one of the intervention groups [17].

Also, flexural eczematous lesions showed greater hardness, less friction and greater surface area, in relation to damaged skin. The surface area refers to the comparison of the size of the patient’s skin surface with a fully stretched skin surface. The increased surface indicates skin alteration in AD. Flexural eczema also presented higher contrast, implying poor skin quality, and higher variance, meaning high roughness [16,18].

Between both sexes, men had greater disease severity (reflected in EASI, IGA and NRS pruritus scales), greater erythema in both healthy skin and flexural eczema and less deformability in healthy skin. Less friction and anisotropy in flexural eczema was also found in male skin. These findings may be related to several factors such as males having more severe AD and females taking more care of their skin by using more sunscreen and moisturisers. A previous study found greater homogeneity and a greater number of wrinkles in the skin of women [19], reflecting higher skin quality or better skin care in women than in men, in agreement with our results.

Regarding age, older patients (≥31) showed greater erythema in flexural eczema, greater hydration in healthy skin, greater friction in healthy skin, greater desquamation in healthy skin and a greater number of corneocytes in healthy skin. In contrast, in those under 31 years of age, higher TEWL was observed in flexural eczema. One of the studies mentioned above found that older patients had higher EASI scores [8] and also correlated with lower skin hydration (SCH) [20]. The higher erythema in older patients might suggest that older patients experience more severe inflammatory responses and the higher desquamation and friction could be linked to age-related changes in skin structure and function. In contrast, younger patients (<31) exhibited higher TEWL in flexural eczema, indicating a greater impairment in skin barrier function in this group. This might suggest that younger patients experience more pronounced moisture loss through the skin, potentially leading to more acute AD flare-ups and an increased need for moisturisers, as reflected by their more frequent use of these products. Thus, while older patients seem to present with more visible skin alterations and a heightened inflammatory response, younger patients appear to have a more compromised skin barrier. These differences highlight the need for age-specific approaches to AD management, addressing the unique clinical manifestations in each age group.

The severity of AD calculated with the EASI, SCORAD and IGA scores showed a relationship with worse subjective disease control by patients with the POEM, DLQI, NRS pruritus and NRS sleep questionnaires. The more severe the disease, the more erythema was observed on healthy skin reflecting the inflammatory phenomenon of the pathology. An increase in pH was found in flexural eczema implying an alkalinisation of the skin and a disruption of this barrier function. The higher the severity was, the lower the R7 elasticity in flexural eczema, the lower the sheen in flexural eczema and the lower the number of flexural eczema cells. Previous studies have linked a higher SCORAD score (and therefore greater severity of AD) to lower SCH and higher TEWL, which, as explained above, is possible due to the presence of mutations or alterations in the number of filaggrins, proteins that make up the stratum corneum, the main barrier function [20,21].

It is also important to mention that the therapeutic paradigm in AD has rapidly shifted and new biologic drugs and JAK inhibitors have emerged that have proven their efficacy in clinical practice [22,23]. However, the patient profile that will respond best to each of them is unknown. Skin barrier function parameters and skin microtopography could help clinicians to select the right drug early in treatment [24].

This is the first study that evaluated skin microtopography in patients with AD and assessed the impact of age, sex and disease severity in these parameters. Nevertheless, it is not exempt from limitations. One of the limitations of our study was the small sample size and the lack of follow-up due to the cross-sectional design of the study. Another limitation is the impossibility of including patients recently diagnosed with AD since most of them were already on treatment, and they were also receiving different treatments, which could influence epidermal barrier function.

## 5. Conclusions

This study demonstrates that barrier function and skin topography are altered in patients with AD, with differences observed between flexural eczema and healthy skin. Flexural eczematous lesions showed greater desquamation, greater smoothness, more wrinkles, greater hardness, less friction, greater surface area, greater contrast, greater variance and fewer cells, reflecting a deterioration in the quality of the skin of these patients. These results may have implications when it comes to establishing a therapeutic plan or influencing the evolutionary course of this pathology.

## Figures and Tables

**Figure 1 jcm-13-05861-f001:**
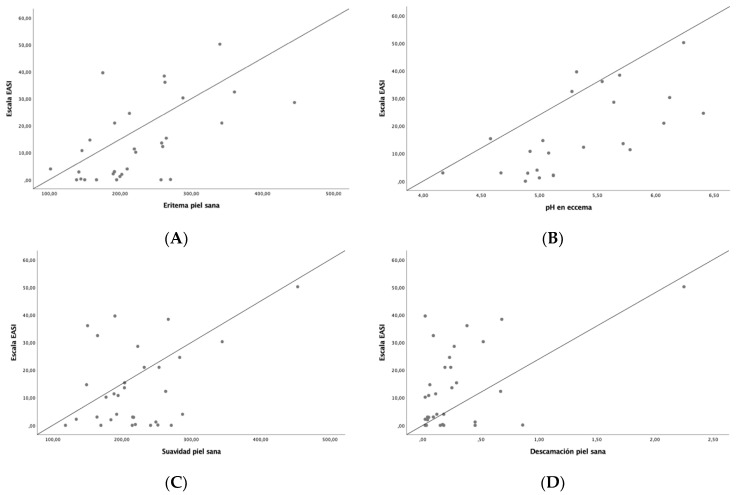
Correlation between EASI and skin barrier function and skin microtopography parameters: (**A**) erythema on healthy skin, (**B**) pH on flexural eczematous lesions, (**C**) smoothness on healthy skin, (**D**) desquamation on healthy skin, (**E**) contrast on healthy skin, (**F**) entropy on healthy skin, (**G**) brightness on flexural eczematous lesions.

**Table 1 jcm-13-05861-t001:** Demographic Characteristics and Descriptive Analysis of the Participants.

Variables	Study Population (n = 32)
Sex	
Male	37.5% (12)
Female	62.5% (20)
Age (years)	30.91 (14.29)
Education Level	
Basic	28.1% (9)
Medium	34.4% (11)
Higher	37.5% (12)
Place of Residence	
Urban	90.6% (29)
Rural	9.4% (3)
Phototype	
I	12.5% (4)
II	43.8% (14)
III	34.4% (11)
IV	9.4% (3)
Smoking (yes)	18.8% (6)
Number of cigarettes/day	1.28 (3.43)
Drinking (yes)	40.6% (13)
Number of SDU per week	1.38 (2.08)
History of skin diseases (yes)	34.4% (11)
Other diseases (yes)	28.1% (9)
Daily medication (yes)	28.1% (9)
Sun exposure (yes)	50% (16)
Time exposed to sun (minutes)	60.94 (85.7)
Use of sunscreen (yes)	50% (16)
Use of moisturiser (yes)	90.6% (29)
Number of times moisturiser is used per week	5.53 (2.36)
Age of onset of AD (years)	11.63 (19.56)
Family members with AD (yes)	43.8% (14)
Associated comorbidities (yes)	46.9% (15)
Asthma (yes)	31.3% (10)
Rhinitis (yes)	37.5% (12)
Conjunctivitis (yes)	28.1% (9)

The values are expressed as mean, standard deviation (SD) or absolute frequencies (%); AD, atopic dermatitis; SDU, standard drinking unit.

**Table 2 jcm-13-05861-t002:** Barrier Function Parameters and Skin Topography in Healthy Skin and Eczema.

Variable	Healthy Skin (n = 32)	Flexural Eczematous Lesions (n = 24)	*p* (Healthy vs. Eczema)
Temperature (°C)	31.81 (1.2)	32.38 (1.11)	0.02
Erythema (AU)	223.89 (74.73)	369.12 (79.41)	<0.001
TEWL (g/h/m^2^)	13.51 (5.57)	27.24 (11.85)	<0.001
SCH (AU)	31.88 (11.56)	20.3 (10.41)	<0.001
pH	5.43 (0.6)	5.32 (0.56)	0.67
Elasticity R0 (%)	0.34 (0.08)	0.34 (0.11)	0.68
Elasticity R2 (%)	0.78 (0.13)	0.78 (0.1)	0.87
Elasticity R7 (%)	0.65 (0.12)	0.56 (0.12)	0.05
Antioxidant capacity (AU)	5.38 (2.65)	4.73 (1.6)	0.45
Hardness (AU)	9.63 (4.68)	15.75 (9.7)	0.02
Friction (AU)	177.84 (114.02)	102.99 (51.46)	0.02
Shine (AU)	6.46 (0.71)	5.94 (1.31)	0.13
Indentometer (mm)	2 (0.4)	1.93 (0.45)	0.77
Smoothness (SEsm)	220.95 (65.32)	316.98 (111.99)	<0.001
Roughness (Ser)	2.21 (1.47)	3.83 (6.06)	0.24
Scaling (SEsc)	0.29 (0.42)	5.57 (7.43)	0.02
Wrinkles (SEw)	62.15 (14.94)	73.33 (26.32)	0.03
Surface (%)	696.31 (84.31)	836.14 (144.54)	<0.001
Volume (mm^2^)	97.1 (14.57)	105.28 (28.42)	0.15
Contrast (AU)	1.31 (0.36)	2.02 (0.88)	0.01
Entropy (AU)	1.48 (0.04)	1.43 (0.04)	<0.001
Variance (AU)	4.96 (0.72)	6.22 (1.25)	<0.001
Homogeneity (AU)	5.28 (22.21)	1.3 (0.08)	0.32
Anisotropy (AU)	30.72 (13.48)	34.33 (17)	0.29
Total number of cells	132.5 (29.31)	105.5 (35.65)	<0.001
Corneofix all (%)	50.35 (16.38)	36.19 (16.54)	0.02
Corneofix all (mm^2^)	14.19 (4.67)	10.42 (4.76)	0.03
Scaling index	26.1 (9.3)	19.36 (10.74)	0.04

*p* value after using Student’s t-test for dependent samples to compare healthy skin and flexural eczematous lesions. AU, arbitrary units.

**Table 3 jcm-13-05861-t003:** Pearson’s Correlation Between Severity Scales of Atopic Dermatitis (AD) and Barrier Function and Topographic Parameters.

Variable	EASI	SCORAD	IGA	NRS Pruritus	NRS Sleep
Healthy Skin Temperature					
Correlation (r)	−0.15	−0.25	−0.16	−0.34	−0.14
*p*	0.43	0.17	0.38	0.06	0.47
Flexural eczema Temperature					
Correlation (r)	−0.3	−0.21	−0.1	−0.2	−0.23
*p*	0.15	0.34	0.36	0.35	0.3
Healthy Skin Erythema					
Correlation (r)	0.56	0.57	0.65	0.33	0.31
*p*	<0.001	<0.001	<0.001	0.07	0.09
Flexural eczema Erythema					
Correlation (r)	0.19	0.23	0.38	0.19	0.3
*p*	0.36	0.29	0.07	0.39	0.16
Healthy Skin TEWL					
Correlation (r)	0.28	0.23	0.19	0.01	0.2
*p*	0.12	0.2	0.3	0.95	0.27
Flexural eczema TEWL					
Correlation (r)	−0.23	−0.18	−0.21	−0.02	−0.18
*p*	0.28	0.39	0.34	0.92	0.4
Healthy Skin SCH					
Correlation (r)	−0.24	−0.3	−0.3	−0.22	0.01
*p*	0.18	0.1	0.1	0.23	0.99
Flexural eczema SCH					
Correlation (r)	−0.1	−0.13	−0.18	0.11	0.12
*p*	0.64	0.55	0.4	0.61	0.58
Healthy Skin pH					
Correlation (r)	0.19	0.23	0.13	0.14	0.12
*p*	0.3	0.21	0.49	0.46	0.53
Flexural eczema pH					
Correlation (r)	0.66	0.73	0.63	0.47	0.4
*p*	<0.001	<0.001	0.001	0.02	0.06
Healthy Skin Elasticity R0					
Correlation (r)	0.29	0.32	0.36	0.23	0.21
*p*	0.11	0.07	0.04	0.22	0.25
Flexural eczema Elasticity R0					
Correlation (r)	0.13	0.18	0.14	0.34	0.28
*p*	0.55	0.39	0.53	0.12	0.2
Healthy Skin Elasticity R2					
Correlation (r)	0.11	0.19	0.15	0.25	0.23
*p*	0.54	0.3	0.42	0.18	0.22
Flexural eczema Elasticity R2					
Correlation (r)	−0.07	−0.2	−0.22	−0.13	−0.32
*p*	0.74	0.36	0.31	0.55	0.13
Healthy Skin Elasticity R7					
Correlation (r)	−0.03	0.06	0.07	0.12	0.16
*p*	0.89	0.74	0.69	0.52	0.39
Flexural eczema Elasticity R7					
Correlation (r)	−0.3	−0.32	−0.42	−0.06	−0.29
*p*	0.15	0.13	0.04	0.78	0.18
Healthy Skin Brightness					
Correlation (r)	−0.27	−0.3	−0.38	−0.19	−0.15
*p*	0.14	0.1	0.03	0.3	0.43
Flexural eczema Brightness					
Correlation (r)	−0.48	−0.48	−0.55	−0.24	−0.25
*p*	0.02	0.02	0.01	0.26	0.24
Smoothness (SEsm) Healthy					
Correlation (r)	0.05	−0.12	0.09	−0.31	0.01
*p*	0.82	0.59	0.7	0.15	0.96
Flexural eczema Smoothness (SEsm)					
Correlation (r)	−0.05	−0.09	−0.09	−0.07	0.04
*p*	0.78	0.62	0.62	0.73	0.85
Roughness (Ser) Healthy					
Correlation (r)	0.05	−0.12	0.09	−0.31	0.01
*p*	0.82	0.59	0.7	0.15	0.96
Flexural eczema Roughness (Ser)					
Correlation (r)	0.33	0.26	0.26	0.07	0.12
*p*	0.11	0.22	0.22	0.74	0.59
Scaling (SEsc) Healthy					
Correlation (r)	0.49	0.31	0.34	0.02	−0.07
*p*	0.04	0.09	0.05	0.9	0.7
Flexural eczema Scaling (SEsc)					
Correlation (r)	0.29	0.22	0.2	−0.1	0.11
*p*	0.16	0.31	0.35	0.62	0.62
Wrinkles (SEw) Healthy					
Correlation (r)	0.2	0.13	0.06	0.11	0.05
*p*	0.28	0.48	0.73	0.56	0.8
Flexural eczema Wrinkles (SEw)					
Correlation (r)	0.37	0.13	0.24	−0.19	−0.16
*p*	0.08	0.55	0.26	0.4	0.47
Healthy Surface					
Correlation (r)	0.32	0.31	0.38	0.12	0.01
*p*	0.07	0.09	0.03	0.52	0.96
Flexural eczema Surface					
Correlation (r)	−0.11	0.02	0.11	0.07	0.43
*p*	0.6	0.92	0.63	0.77	0.04
Healthy Contrast					
Correlation (r)	0.37	0.34	0.41	0.14	−0.01
*p*	0.04	0.06	0.02	0.45	0.98
Flexural eczema Contrast					
Correlation (r)	−0.08	0.06	0.12	−0.03	0.32
*p*	0.72	0.8	0.59	0.91	0.14
Healthy Entropy					
Correlation (r)	−0.42	−0.38	−0.42	−0.18	−0.09
*p*	0.02	0.03	0.02	0.32	0.63
Flexural eczema Entropy					
Correlation (r)	−0.003	−0.06	−0.19	−0.01	−0.32
*p*	0.99	0.79	0.37	0.99	0.13
Healthy Variance					
Correlation (r)	0.32	0.31	0.39	0.12	0.01
*p*	0.07	0.09	0.03	0.51	0.95
Eczema Variance					
Correlation (r)	−0.1	0.03	0.11	0.05	0.43
*p*	0.64	0.89	0.61	0.83	0.04
Healthy Homogeneity					
Correlation (r)	0.46	0.27	0.26	−0.01	−0.07
*p*	0.01	0.13	0.16	0.99	0.7
Flexural eczema Homogeneity					
Correlation (r)	0.22	0.05	−0.01	−0.09	−0.25
*p*	0.31	0.83	0.99	0.7	0.26
Total Number of Cells Healthy					
Correlation (r)	−0.01	−0.02	0.15	−0.01	0.13
*p*	0.95	0.93	0.42	0.98	0.48
Flexural eczema Total Number of Cells					
Correlation (r)	−0.42	−0.32	−0.32	−0.05	−0.1
*p*	0.04	0.12	0.13	0.83	0.66
Corneofix all % Healthy					
Correlation (r)	0.03	−0.09	−0.22	−0.1	−0.05
*p*	0.86	0.62	0.22	0.6	0.81
Flexural eczema Corneofix all %					
Correlation (r)	0.23	0.21	0.12	0.37	0.34
*p*	0.29	0.32	0.57	0.08	0.11
Corneofix all mm^2^ Healthy					
Correlation (r)	−0.14	−0.2	−0.32	−0.1	−0.02
*p*	0.44	0.28	0.07	0.6	0.93
Flexural eczema Corneofix all mm^2^					
Correlation (r)	0.23	0.21	0.12	0.37	0.34
*p*	0.29	0.32	0.57	0.08	0.11
Scaling Index Healthy					
Correlation (r)	0.05	−0.07	−0.2	−0.08	−0.05
*p*	0.77	0.7	0.29	0.68	0.8
Flexural eczema Scaling Index					
Correlation (r)	0.22	0.21	0.14	0.37	0.4
*p*	0.29	0.31	0.52	0.08	0.06

r = Pearson’s correlation, *p* = value of p after Pearson’s correlation (r).

## Data Availability

The data supporting the reported results are available on request from the corresponding author. The data are not publicly available due to ethical restrictions.

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
