# Peer review of "Skin Barrier Function and Microtopography in Patients with Atopic Dermatitis"

_jcm, 2024, doi:10.3390/jcm13195861_

Round 1

Reviewer 1 Report

Comments and Suggestions for Authors

The work by pretel-lara et al. is comprehensive, clear and interesting. I have not major issues.

I suggest a more comprehensive discussion on novel therapy for atopic dermatitis, I suggest some references: Mastorino L, Cantafio Duò VL, Vecco C, et al. Impact of comorbidities in the response of atopic patients treated with dupilumab: a real-life study up to 36 weeks. J Eur Acad Dermatol Venereol. 2022;36(12):e1021-e1023. doi:10.1111/jdv.18427

and De Greef A, Ghislain PD, de Montjoye L, Baeck M. Real-Life Effectiveness and Tolerance of Upadacitinib for Severe Atopic Dermatitis in Adolescents and Adults. Adv Ther. 2023;40(5):2509-2514. doi:10.1007/s12325-023-02490-5

Author Response

The work by pretel-lara et al. is comprehensive, clear and interesting. I have not major issues.

Thank you very much for all your comments.

I suggest a more comprehensive discussion on novel therapy for atopic dermatitis, I suggest some references: Mastorino L, Cantafio Duò VL, Vecco C, et al. Impact of comorbidities in the response of atopic patients treated with dupilumab: a real-life study up to 36 weeks. J Eur Acad Dermatol Venereol. 2022;36(12):e1021-e1023. doi:10.1111/jdv.18427

and De Greef A, Ghislain PD, de Montjoye L, Baeck M. Real-Life Effectiveness and Tolerance of Upadacitinib for Severe Atopic Dermatitis in Adolescents and Adults. Adv Ther. 2023;40(5):2509-2514. doi:10.1007/s12325-023-02490-5

Thank you for the suggestion. We have included a comprehensive discussion on novel therapies for atopic dermatitis and have included the references suggested. The following paragraph have been included in the discussion: It is also important to mention that therapeutic paradigm in AD has rapidly shifted and new biologic drugs and JAK inhibitors have emerged that have proven their efficacy in clinical practice22,23. However, the patient profile that will respond best to each of them is unknown. Skin barrier function parameters and skin microtopography could help clinicians to select the right drug early24.

Reviewer 2 Report

Comments and Suggestions for Authors

Dear Authors,

thanks very much for your research. AD is a disease still to be discovered in some aspects and your topic is very interesting and is a novelty.

Introduction is balanced and topic-focused.

Materials and methods are well described particularly in explanation of instruments used for parameters evaluation.

Results are clear and exhaustive, more detailed. I found more interesting the differences beetwen sex and age. 

It would be very interesting to evaluate these parameters during therapy (before and after) and between the various therapies.  Do you have any preliminary data on this matter?

Discussion and Conclusions are adequate with many references to the literature and with a interesting discussion about study limitations.

I suggest to uniform the table 2 with other table.

I suggest to report the city near the hospital name.

Line 232 in is repeated twice.

Line 328-29 there is a repetition microtopography topography.

Author Response

Dear Authors,

thanks very much for your research. AD is a disease still to be discovered in some aspects and your topic is very interesting and is a novelty.

Thank you for your comments

Introduction is balanced and topic-focused.

Materials and methods are well described particularly in explanation of instruments used for parameters evaluation.

Results are clear and exhaustive, more detailed. I found more interesting the differences beetwen sex and age. 

Thank you for your comments and reviewing the manuscript.

It would be very interesting to evaluate these parameters during therapy (before and after) and between the various therapies.  Do you have any preliminary data on this matter?

We are working now in this aspect. We haven’t collected many patients for now but we have observed that dupilumab could improve some parameters such as wrinkles, smoothness, roughness and flaking, related to an improvement in skin microtopography.

Discussion and Conclusions are adequate with many references to the literature and with a interesting discussion about study limitations.

Thank you very much for all the comments. 

I suggest to uniform the table 2 with other table.

Table 2 has been uniformed.

I suggest to report the city near the hospital name.

The city has been reported near the hospital name.

Line 232 in is repeated twice.

Thank you. One in has been omitted

Line 328-29 there is a repetition microtopography topography.

“Topography! Has been omitted.

Reviewer 3 Report

Comments and Suggestions for Authors

1.The number of patients is quite small for a disease with a high prevalence. Maybe it was good to separate them by age groups, respectively12-18 and over 18. They are different skin types,different therapeutic indications a.t.o.

2.It needed to be better explained why the age of 31 years separating the two groups of patients was chosen.

3.The differences between the parameters evaluated in the two ages categories are not very clearly interpreted.

Author Response

1.The number of patients is quite small for a disease with a high prevalence. Maybe it was good to separate them by age groups, respectively12-18 and over 18. They are different skin types,different therapeutic indications a.t.o.

Thank you for the comments. We agree that one of the limitations is the limited sample size so it has been mentioned in the study limitations. I would be interesting to conduct further studies with a great number of patients and evaluate the impact of other factors as recommended included skin types or drugs

2.It needed to be better explained why the age of 31 years separating the two groups of patients was chosen.

The age of 31 years represents the mean age of the study population. This threshold was used to separate the two groups for comparison purposes, reflecting a midpoint that helps in distinguishing between younger and older participants. A more detailed explanation of why this particular age was chosen can clarify that it provides a balanced division based on the population's demographic distribution, allowing for meaningful analysis of age-related differences in the parameters evaluated. This information has been added in the methods to clarify this aspect.

3.The differences between the parameters evaluated in the two ages categories are not very clearly interpreted.

Thank you for this interesting issue. To provide a clearer interpretation of the differences between the two age categories, it's important to contextualize the findings based on age-related physiological changes and clinical relevance. The results show that patients aged ≥31 exhibited greater erythema in eczema, higher SCH, increased friction, and desquamation in healthy skin. The higher erythema in older patients might suggest that older patients experience more severe inflammatory responses and the higher desquamation and friction could be linked to age-related changes in skin structure and function. In contrast, younger patients (<31) exhibited higher TEWL in eczema, indicating a greater impairment in skin barrier function in this group. This might suggest that younger patients experience more pronounced moisture loss through the skin, potentially leading to more acute eczema flare-ups and an increased need for moisturizers, as reflected by their more frequent use of these products. Thus, while older patients seem to present with more visible skin alterations and a heightened inflammatory response, younger patients appear to have a more compromised skin barrier. These differences highlight the need for age-specific approaches to AD management, addressing the unique clinical manifestations in each age group. This information has been added in the discussion.